# Vibrational fingerprint of localized excitons in a two-dimensional metal-organic crystal

M. Corva[1,2], A. Ferrari[1,5], M. Rinaldi[1], Z. Feng [1,6], M. Roiaz[3], C. Rameshan[3], G. Rupprechter [3], R. Costantini[1,2], M. Dell'Angela[2], G. Pastore[1], G. Comelli[1,2], N. Seriani[4] & E. Vesselli [1,2]

Long-lived excitons formed upon visible light absorption play an essential role in photovoltaics, photocatalysis, and even in high-density information storage. Here, we describe a self-assembled two-dimensional metal-organic crystal, composed of graphene-supported macrocycles, each hosting a single $FeN_4$ center, where a single carbon monoxide molecule can adsorb. In this heme-like biomimetic model system, excitons are generated by visible laser light upon a spin transition associated with the layer 2D crystallinity, and are simultaneously detected via the carbon monoxide ligand stretching mode at room temperature and near-ambient pressure. The proposed mechanism is supported by the results of infrared and time-resolved pump-probe spectroscopies, and by ab initio theoretical methods, opening a path towards the handling of exciton dynamics on 2D biomimetic crystals.

[1] Dipartimento di Fisica, Università degli Studi di Trieste, via A. Valerio 2, Trieste 34127, Italy. [2] Istituto Officina dei Materiali CNR-IOM, S.S. 14 km 163.5, Area Science Park, Basovizza, Trieste 34149, Italy. [3] Institute of Materials Chemistry, Technische Universität Wien, Getreidemarkt 9-BC-01, Vienna A-1060, Austria. [4] The Abdus Salam International Centre for Theoretical Physics, Str. Costiera 11, Trieste 34151, Italy. [5] Present address: ICAMS, Ruhr-Universität Bochum, Universitätsstr. 150, Bochum 44801, Germany. [6] Present address: Department of Chemical Engineering and Materials Science, University of California, Irvine, Irvine, CA 92697, USA. Correspondence and requests for materials should be addressed to E.V. (email: evesselli@units.it)

The absorption of visible light in a material can lead to the formation of excitons, i.e. pairs composed of an excited electron and the associated hole. Nature exploits excitons in nanoscale systems in order to harvest and funnel energy to chemical reaction centers. Their properties and dynamics are therefore crucial for biomimetic optoelectronic, photovoltaic, and photocatalytic applications, but excitons are difficult to control and characterize. In particular, reducing electron–hole pair recombination is essential to produce long-lived excitons, maximizing efficiency.

Long-lived excitons in molecular systems have been observed in recent years, with lifetimes up to hundreds of picoseconds at room temperature[1]. The mechanism determining the coherence time is based on the coupling between the exciton and the bath of surrounding nuclear motions, in connection with self-trapping localization effects. Indeed, the soft character of organic frameworks induces a strong coupling between molecular, nuclear, and electronic dynamics[2], resulting in a vibronic manifold with high density of states that can facilitate mixing[3]. Another important contribution to the exciton lifetime originates from the occurrence of spin transitions, yielding spin crossover metastable states. Indeed, if a switch in the spin configuration of the molecule occurs, the system can be trapped in a triplet excited, none-quilibrium state. The spin transition can be driven by symmetry breaking and entropy differences[4], associated with spin multiplicity and with the number of accessible vibrational states[5]. In several iron(II) complexes, spin transitions can be induced by thermal excitation, but light-induced excited spin state trapping (LIESST) through a multiple-step mechanism has also been observed[6,7], thus enabling the potential development of light-controlled molecular bits in spintronic devices. The search for materials capable of trapping excited states even at room temperature has started long ago, and some promising systems have already been identified[6,8]. The role of ligand fields in the light-induced spin change process was also discussed[7,8]. When light is exploited to induce the singlet–triplet transition as in the LIESST process, the triplet generation process may be associated with intersystem crossing or singlet fission mechanisms, thus involving a short-lived, intermediate singlet excited state that relaxes through a cooperative path involving adjacent chromophores[9]. In the latter case, light absorption generates a spin-singlet exciton that converts within few hundreds of femtoseconds into a pair of spin-triplet excitons residing at neighboring sites[3]. The two generated spin-triplet excitons initially correlate to form an overall spin-singlet state, thus making the transition spin-allowed. Singlet fission does therefore not occur in single, isolated small-molecules, since at least two adjacent sites are needed to accommodate the triplet excitation products[9]. Finally, for singlet fission to be spontaneous, the singlet excitation energy needs to be more than twice the triplet excitation energy. So far, these processes have been observed only in nanocrystals or thin films, but not in a purely 2D system to the best of our knowledge. Exciton multiplication by singlet fission could indeed yield practical applications by improving the efficiency of photovoltaic devices[4,9], as well as the adsorption properties of gas sequestration and conversion devices by exploiting cooperative mechanisms[10].

Here, we demonstrate that, by means of ligand adsorption, it is possible to localize long-lived Frenkel excitons in a self-assembled 2D molecular system, and to detect them via their effect on the vibrational spectrum of the ligand itself (carbon monoxide), even at room temperature and near-ambient pressure. We show that nonlinear optical sum-frequency generation (SFG) spectroscopy is a unique tool for this kind of investigation as it mimics a pump-and-probe experiment, paving the way for simultaneous controlled generation and characterization of localized excitons in a 2D metal–organic system on a laboratory scale. Moreover, the generation of long-lived excitons is affected by the degree of order of the metal–organic monolayer due to the quantum-mechanical nature of the involved processes, suggesting a practical approach to systematically tune the excitonic properties. We describe a self-assembled 2D metal–organic crystal, composed of graphene-supported macrocycles. Each of the latter host a FeN$_4$ center at which a CO ligand can bind. This system represents a heme-like biomimetic model, where we generate excitons by visible laser light, and detect them by observing the influence on the CO–ligand stretching mode at room temperature and near-ambient pressure. These findings may have an impact on the fundamental understanding of exciton "writing and reading" and could open up pathways toward more efficient photovoltaic, photocatalytic, or information storage devices.

## Results

**Sample characterization.** It would be of fundamental and technological interest to mimic the above electron- and spin-transition properties within the framework of controlled and synthetic (model) systems. For this purpose, we exploit metal–organic 2D films and crystals[11], which can be obtained by self-assembly methods on a weakly interacting and convenient support like graphene, yielding a π–conjugated system. For triplet exciton formation to take place, it is crucial that the inter-molecular coupling is on the one hand strong enough to ensure efficient singlet fission and on the other hand weak enough to allow fast diffusion and separation of triplets, providing optimal conditions[9]. Singlet excitons are short-lived and might be delocalized over more than one site in a 2D crystal, whereas triplet excitons are long-lived, and tend to be localized on a single molecular site[9]. The ability of excitons to diffuse reduces the recombination chances, making molecular crystals efficient systems for singlet fission processes[9]. In the specific case, we have chosen to adsorb Fe-phthalocyanines (FePc) under ultra-high vacuum (UHV) conditions on a single, complete graphene (GR) sheet grown in situ on the Ir(111) single crystal termination (Fig. 1a). The FePc molecules are thermally stable[12–14], weakly couple to the GR support[15], and form a well-ordered 2D crystal with almost square symmetry (Fig. 1b, c). We exposed this system to a near to atmospheric pressure (1–10 mbar) of carbon monoxide. By exploiting IR-Vis Sum-Frequency Generation spectroscopy (IR-Vis SFG), we monitored the evolution of the C–O stretching intensity as a function of the CO pressure at room temperature, thus following the FePc carbonylation process in situ. In the following, we will show that IR-Vis SFG acts as a pump-and-probe experiment, generating long-lived excitons, which are detected by their effect on the vibrational modes of the CO ligand. This approach offers considerable advantages over other methods, being able to investigate the system in situ at near-atmospheric conditions[16,17]. In fact, phthalocyanines have already shown nonequilibrium spin crossover properties that can be locally controlled by means of the tip of a scanning tunneling microscope (STM) after adsorption at cryogenic temperature in UHV on a substrate that crucially affects the spin state of the central metal atom[18–20]. For similar systems, STM has also shown the potential to observe the dynamic behavior of organic molecules at surfaces, yielding detailed information about energetics and kinetics[21]. As an alternative, adsorption and thermal desorption of small ligand molecules that interact with the fourfold coordinated metal center were employed to tune the magnetic coupling of metal–organic complexes to substrates[22–24]. The light-induced switching can be exploited as a way to generate a nonequilibrium or metastable molecular electronic and spin configuration, since the optical pumping yields excitation of the

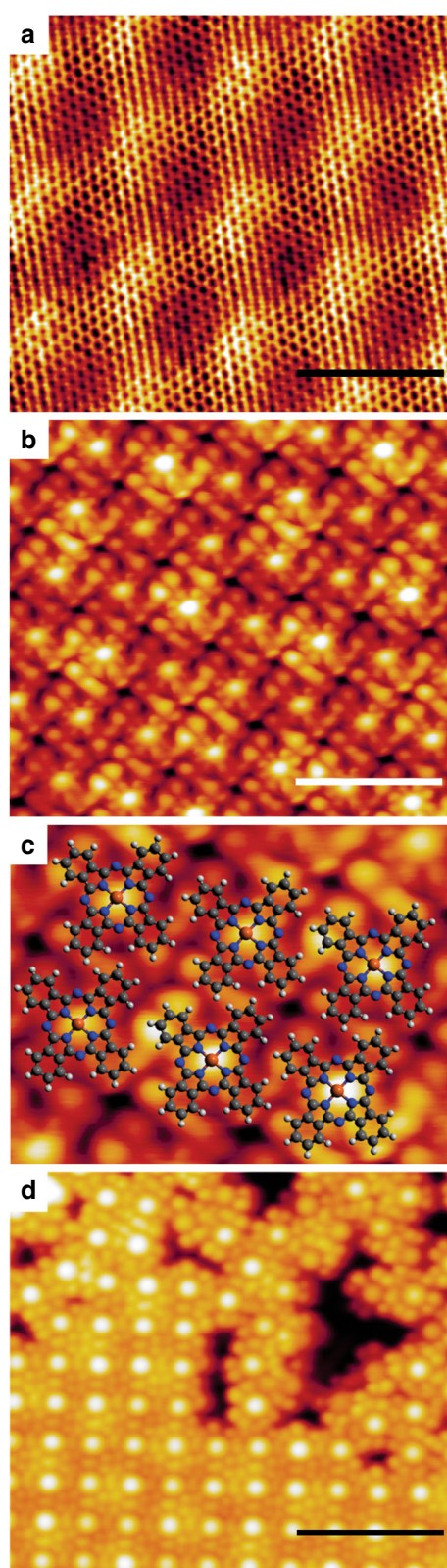

**Fig. 1** Structural characterization of the FePc/GR system. **a** STM image of bare graphene on Ir(111). **b** STM image of the FePc single, well-ordered layer on graphene and **c** the structural model of the molecules. **d** A boundary between ordered and disordered domains. All images were taken under UHV conditions at 4 K [**a** $V_{bias} = +0.1$ V, $I = 1$ nA; **b, c** $V_{bias} = -2.0$ V, $I = 0.2$ nA; **d** $V_{bias} = -2.0$ V, $I = 0.1$ nA; error bars: **a**, **b** 3 nm, **d** 5 nm]

metal-to-ligand charge transfer that rapidly decays to a high-spin state[5]. However, cryogenic temperatures are needed to promote ligand adsorption and immobilization of the molecules, quenching the spin properties originating from the interaction with the supporting, underlying substrate[5].

**Ligand adsorption**. We have grown the ordered FePc monolayer under UHV conditions on the single, complete GR sheet on the Ir (111) single crystal termination. The almost square lattice, characterized by an angle of 87° (Fig. 1b, c), similar to the case of CoPcs[25], has lattice vector lengths of 14.1 and 13.6 Å. The FePc monolayer lattice is rotated by 13° with respect to the underlying GR sheet, while the molecules are rotated by 18° with respect to the FePc lattice. Depending on the exact procedure (FePc deposition and post-annealing temperature), the prepared layers may show the co-existence of both ordered and disordered domains (Fig. 1d). While following the carbonylation process in situ, we observed the growth of four distinct vibronic features in the C–O stretching region for the 2D FePc/GR crystal, while no such features were observed for the multilayer, as detailed below. A complete bare GR sheet is not active toward CO adsorption and passivates the underlying Ir metal surface[26,27]. Indeed, upon exposure of the bare GR/Ir(111) system to several mbar of CO, IR-Vis SFG still detects only the GR phonon at 1608 cm$^{-1}$, corresponding to the in-plane optical G modes associated with the stretching of the pairs of $sp^2$-bonded C atoms[28], while no C–O stretching is observed (Supplementary Fig. 1). Intercalation of CO under the GR sheet would produce a very intense vibronic signal in the 2040–2080 cm$^{-1}$ range[26]. Instead, on the 2D FePc/GR/Ir (111) crystal features associated with the C–O stretch progressively develop in the 2000 cm$^{-1}$ region for CO pressure above 1 mbar at room temperature. Due to the large distance between Fe centers at which the CO molecules bind, a contribution of coverage-dependent nonlinear dipole effects to the SFG intensity can be ruled out, indicating that the observed IR-Vis SFG signal can be exploited to measure the CO uptake on the Fe centers (Fig. 2). By fitting the experimental data with an established Langmuir adsorption kinetic model (dashed line), we obtain a CO–Fe binding energy of $-0.3 \pm 0.1$ eV. In parallel, we performed a thorough set of ab initio calculations in order to get a deeper understanding of the system. Concerning the CO adsorption energy, we obtain a theoretical value of $-0.34$ eV for a single CO ligand adsorbed at each iron center (mono-coordination), in perfect agreement with the experimental data, and find that further CO adsorption beyond mono-coordination is strongly endothermic. The experimental uptake profile is very similar to that reported for FePcs adsorbed at the bare Ir metal surface[29], and the saturation pressure at room temperature is in line with working gas partial pressures of heme systems in nature[30]. In addition, by means of ab initio thermodynamics we computed the (CO)FePcs saturation curve (blue line in Fig. 2). The remarkable agreement further confirms the proper description of the CO bonding mechanism (see Supplementary Note 1). We also find from our calculations that the carbonylation of the FePc quenches the spin of the molecule (see Supplementary Fig. 2 for further details), i.e. the total spin of the (CO)FePc is $S = 0$, at variance with the FePc molecule without the CO ligand ($S = 1$), in agreement with the literature[31]. Therefore, the expected ground state is a singlet state (paired electrons).

**Vibrational characterization and the role of visible light**. A detailed analysis of the IR-Vis SFG C–O stretch fingerprint (Fig. 3 and Supplementary Figs. 3–6) identifies four contributions at 1986, 1992, 2005, and 2011 cm$^{-1}$. The SFG signal was deconvoluted by means of an established least-square fitting

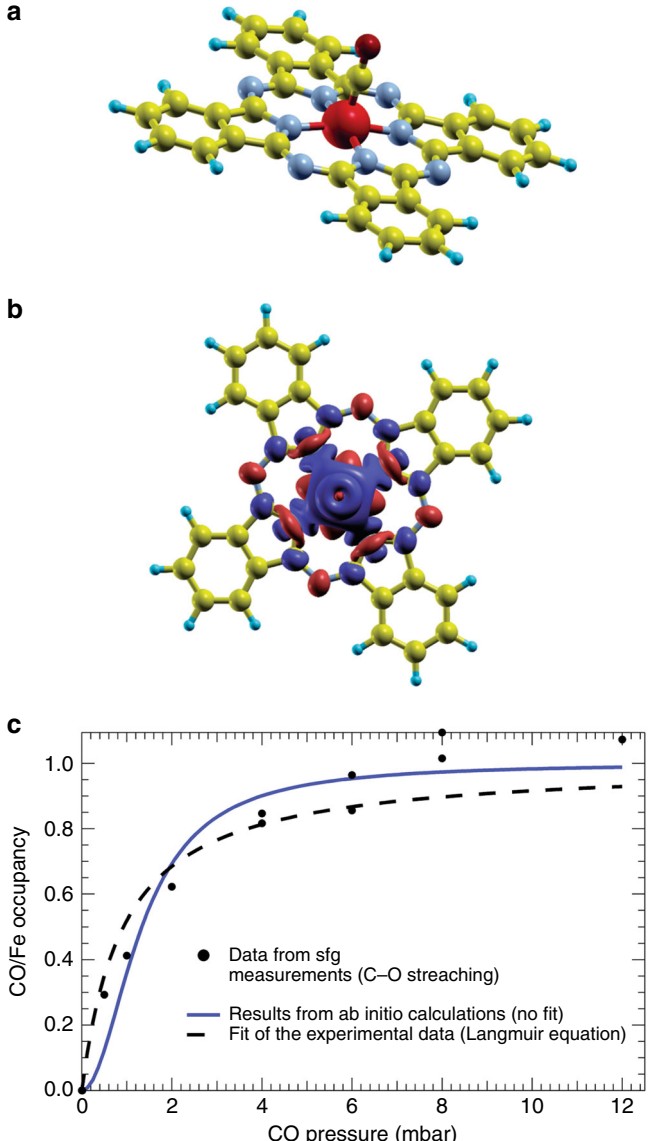

**Fig. 2** Carbonylation of the FePc monolayer on graphene. **a** Adsorption geometry for a single CO molecule on FePc. **b** Spin polarization ($\rho_{up}$–$\rho_{down}$) for the triplet obtained in DFT + U (blue: negative values; red: positive values). **c** Relative coverage of the adsorption sites of a layer of FePc molecules as a function of the CO pressure. The experimental data (dots) are affected by an error of roughly ± 0.1 on the vertical axis. The theoretical curve (blue, solid line) is obtained within the framework of a pure ab initio approach, while the experimental data are fitted by a Langmuir isotherm (black, dashed line)

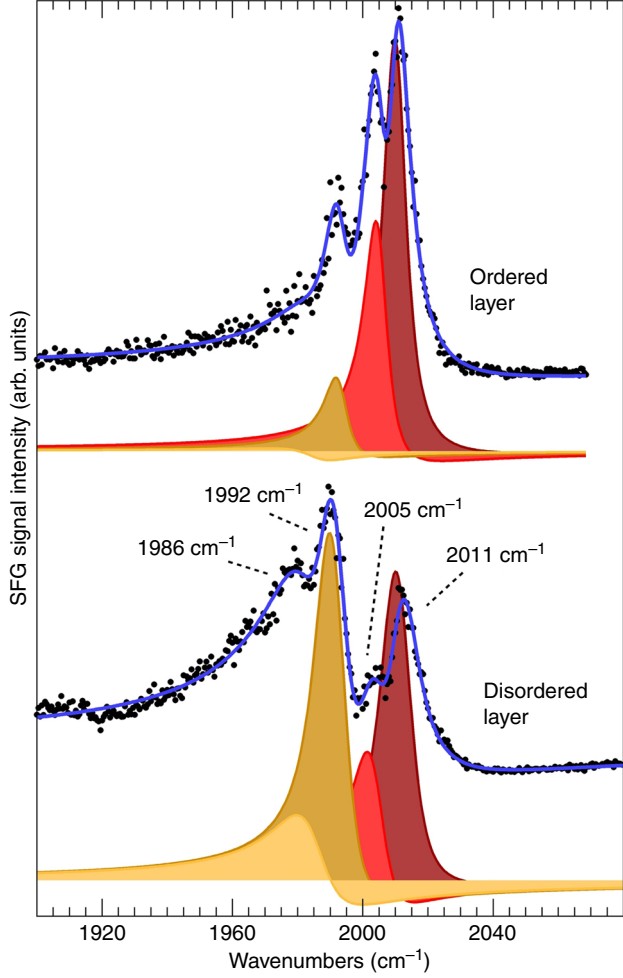

**Fig. 3** IR-Vis SFG intensity spectra. Spectra of the C–O stretching region collected in situ at room temperature in 10 mbar CO on the ordered (top) and disordered (bottom) FePc monolayer on graphene. Experimental data (black dots) are represented, together with the results of the best fit (blue lines) and the interference of each resonance with the nonresonant background according to the effective susceptibility, as described in the text ($\lambda_{Vis} = 532$ nm; ppp polarization)

procedure[32], according to an effective description of the non-linear second-order susceptibility (the complete set of fitting parameters is reported in Supplementary Tables 1 and 2). The surface density of the Fe centers is very low due to the large size of the FePc molecules (about 0.03 ML with respect to the underlying Ir metal surface). Despite this, we managed to measure the resonant signal due to the strong CO dipole and the graphene-enhanced scattering. In the present case, we have used $\lambda_{Vis} = 532$ nm (photon energy: 2.33 eV), thus matching a typical singlet excitation energy[9]. The degree of order of the layer strongly affects the resonant lineshape, as can be observed from the two spectra plotted in Fig. 3, showing a remarkable change of the relative peaks intensities and a Gaussian broadening for the disordered system (see Supplementary Tables 1 and 2), while all

other lineshape parameters including line positions remain constant. In contrast, carbonylation of a FePc multilayer (3D crystal) adsorbed on the same graphene sheet yields a single resonant peak at 2012 cm$^{-1}$ with a different relative phase with respect to the nonresonant background (Fig. 4a and b and Supplementary Table 1). The multiplet is therefore an inherent feature of the 2D system. Analogous measurements were also performed by means of Polarization–Modulation Infrared Reflection Absorption Spectroscopy (PM-IRAS), for which a broadband mid-IR radiation beam is shined on the sample surface[33], thus inducing pure vibrational transitions instead of vibronic excitations. The PM-IRAS spectra obtained upon room temperature carbonylation of the FePc multi- and mono-layers on GR are reported in Fig. 4c and d, respectively. At variance with the corresponding vibronic (SFG) spectra, only one peak appears in both cases, at 2008–2009 cm$^{-1}$. This clearly associates the vibronic splitting observed by IR-Vis SFG with the electronic transitions induced by the impinging visible light. The small experimental discrepancy in the vibrational energies (2–4 cm$^{-1}$) between SFG and IR experiments is likely due to the anharmonic coupling of the C–O stretch to frustrated lateral modes[34].

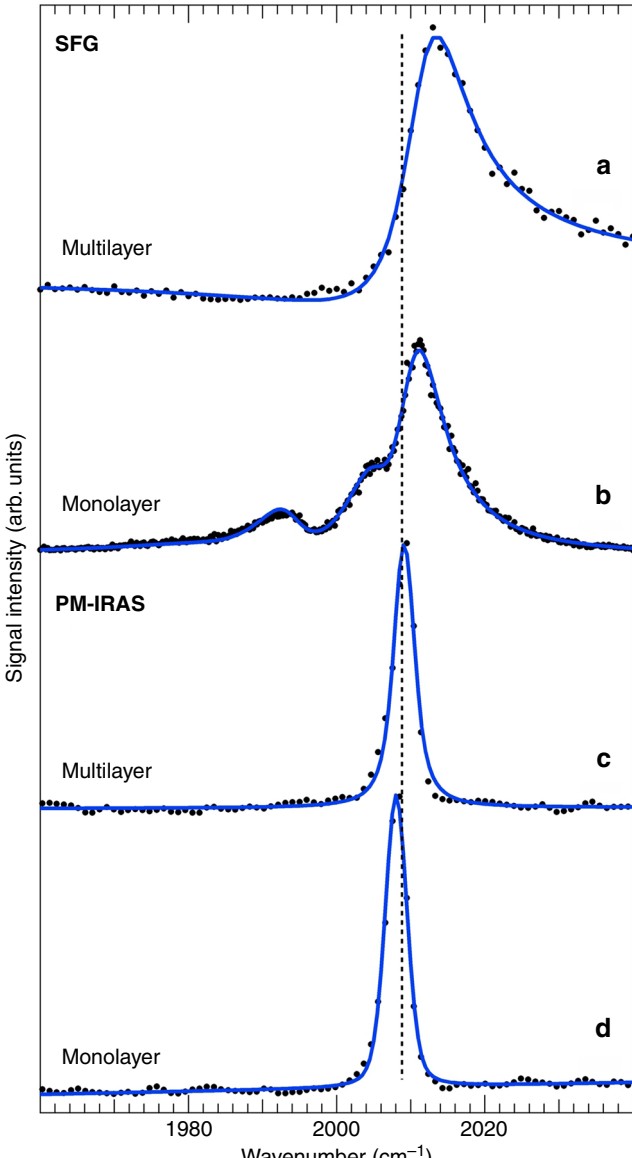

**Fig. 4** Vibrational vs. vibronic fingerprints of the C–O stretching mode for the carbonylated FePc monolayer. Comparison between IR-Vis SFG (**a** and **b**) and PM-IRAS (**c** and **d**) intensity spectra collected in situ at room temperature in 10 mbar CO in the C–O stretching region for the carbonylated FePc monolayer (**b** and **d**) and multilayer (**a** and **c**) on graphene. For the IR-Vis SFG spectra, $\lambda_{Vis} = 532$ nm and ppp polarization were used

**Exciton dynamics**. In the following, we discuss possible scenarios of interpretation. Trans-coordination and ligand environment effects in heme carbonyls have been found to induce shifts in the C–O stretching mode (of the same order as the ones we measure: 5–10 cm$^{-1}$), both in model and in biologic systems[35–37]. A varying coordination of the Fe atom to the underlying graphene sheet and an inhomogeneous coupling contribution[38,39], depending on non-equivalent adsorption geometries of the FePc molecules with respect to the graphene Moiré, cannot account for the observed multiplet since it is absent in pure IR measurements. The measured splitting can neither be ascribed to multiple CO coordination, since the symmetric and antisymmetric stretching modes in Fe poly-carbonyls would be separated by several tens of wavenumbers, at variance with the observed splitting of only a few cm$^{-1}$. Consistently, the absence of the splitting in our pure IR

spectra excludes these interpretations. Finally, as mentioned, our DFT calculations confirm that only mono-coordination is energetically possible.

In order to investigate the actual lifetime of the excited electronic states, we studied the electron dynamics of the FePc monolayer on GR/Ir(111) by means of Time-Resolved Two-Photon Photoemission (TR-2PPE). The experiments were performed at room temperature in order to correlate with the IR-Vis SFG results. However, due to the nature of the photoemission process involving low kinetic energy photoelectrons, we investigated the system in UHV conditions, thus without adsorbed CO. We employed a 515 nm pump pulse to create a transient population of excitonic states that were subsequently probed by a 257.5 nm pulse[40]. Two transient states were detected in our measurements (see Supplementary Note 1 and Supplementary Fig. 7). A short-lived excited state (SL– $\tau_{SL} < $ 570 fs), located 1.3 eV above the Fermi level, was found to decay in a long-lived, lower energy state (LL – $\tau_{LL} = 28 \pm 8$ ps) at E-$E_F \leq 0.6$ eV. We associate the SL state with the electrons excited in the FePc LUMO, supported by previous STS measurements in literature[41]. A precise determination of the energy position of the LL state was not possible, because the first 0.5 eV above the secondary electron cutoff were not accessible with our current experimental setup due to detector saturation. Nevertheless, an upper binding energy limit of 0.6 eV could be obtained. Our findings on this system are in agreement with previous four wave mixing measurements performed on FePcs in a dimethylsulfoxide solution, yielding a short-lived state ($\tau < 170$ fs) decaying in a longer lifetime state ($\tau \sim 40$ ps)[42], and with other pump–probe investigations[43].

## Discussion

Herein, we propose that the observed SFG vibronic splitting is associated with contributions from excited triplet electronic molecular states (unpaired electrons) that are populated upon exposure to the visible light beam, similar to a LIESST scenario. We calculated the vibrational modes for the singlet ground state and for two different triplet states, (see Supplementary Fig. 2), and obtained vibrational frequencies of 2016, 2049, and 2055 cm$^{-1}$, respectively. The calculated splitting is comparable with the splitting among peaks in the SFG spectrum, spanning about 30 cm$^{-1}$ and with a separation between adjacent features of 6 cm$^{-1}$. The increase in the C–O stretching frequency calculated for the triplet states with respect to the ground configuration is consistent with the fact that CO adsorption is weakened, in agreement with an electron back-donation mechanism according to the Blyholder model of CO bonding, making this molecule an ideal probe of the local electron density. A number of observed non-equivalent resonances agrees well with a level splitting that is explained within the framework of a crystal ligand field theory picture, in which a C$_{4v}$ symmetry system is expected to split its energy levels. The latter geometry applies indeed to the carbonylated FePc, since binding of CO lifts the Fe atom from the molecular plane, inducing a symmetry break that activates the singlet fission process, as recently observed[4]. Indeed, the dependence of the relative intensity of the peaks on the ordering of the 2D crystal can be associated with quantum cooperative interactions between the interacting molecules within the framework of a singlet fission origin of the triplet excited states, allowing for subsequent triplet diffusion and minimizing recombination[9]. Accordingly, it is known that there are three synthetic strategies to create and/or strengthen cooperativity, i.e. incorporation of hydrogen-bonded networks, incorporation of π-stacking, and coordination of ligands.[7] Moreover, low- to high-spin conversions are accompanied by profound changes in the molecular

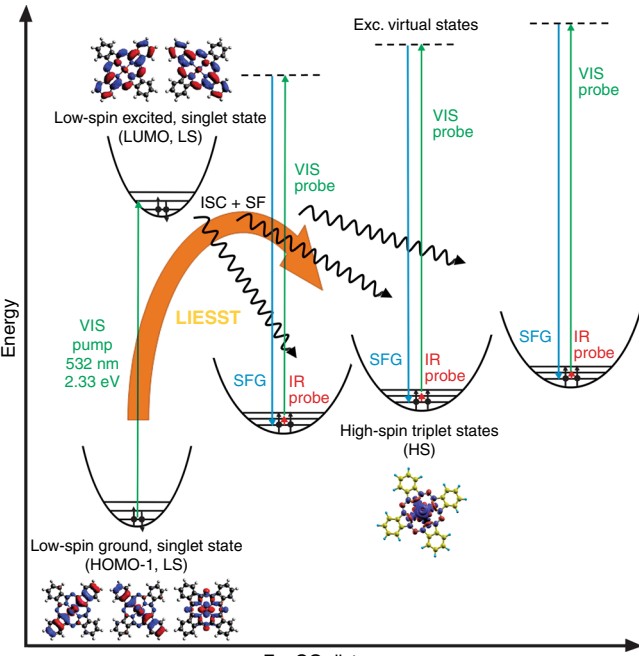

**Fig. 5** Jablonski diagram of the exciton formation mechanisms. The visible radiation yields excitation of the fundamental low-spin (LS) state into an excited singlet configuration, followed by fast relaxation into long-lived triplet high-spin (HS) states through inter-system crossing (ISC) and singlet fission (SF) mechanisms

properties, depending on the new distribution of the $3d$ valence electrons, and therefore including optical, magnetic, structural, and indeed vibrational fingerprints[7]. In the present case, the CO molecule adsorbed at the Fe center acts both as a ligand and as a vibrational identifier. Corroborated by our ab initio DFT simulations, we can propose the mechanism depicted in the Jablonski diagram of Fig. 5. The visible radiation pumps the system from a low-spin ground, singlet state ($s = 0$) to an excited singlet state (HOMO-1 to LUMO transition). The latter cooperatively decays via a singlet fission process to high-spin triplet states that can then be probed by means of IR-Vis SFG. Time-resolved 2PPE experiments (see Supplementary Fig. 7 and Supplementary Note 1) yield lifetimes of less than 570 fs (SL) and of $28 \pm 8$ ps (LL), respectively, for the two corresponding excited states of the decarbonylated FePc/GR, in line with results obtained on similar molecules[40,44–47]. This explains why the SFG measurements are able to catch the population of the most stable configuration (LL), since the IR and Vis pulses have a similar duration. Moreover, we find energies of 1.3 and < 0.6 eV for the two excited states, respectively. This further supports our findings since, as observed in the introduction, for singlet fission to be spontaneous, the singlet excitation energy needs to be more than twice the triplet excitation energy[9].

In summary, we have shown that a self-assembled carbonylated metal–organic 2D crystal supported on graphene, upon exposure to visible light at room temperature, can sustain long-lived excitons (tens of ps). Both the presence of the CO ligand at the Fe center and the 2D network are essential for the stabilization of the excitons that is why they are not observed on the multilayer. This is due to the peculiar mechanisms involved in the LIESST process, including singlet fission that relies on the appropriate interaction between neighboring centers. A templating effect by the substrate may also be further exploited. Moreover, this work shows that IR-Vis SFG is a unique tool for this kind of investigation as it mimics a pump-and-probe experiment. Visible radiation generates

excitons that can be simultaneously detected through their effect on the internal stretching mode of the adsorbed carbon monoxide ligand, a convenient and accurate way to control, investigate, and characterize complex processes in metal–organic molecules, in which electronic and nuclear degrees of freedom interact in non-trivial ways.

## Methods

**Sample preparation.** The same sample preparation recipes were applied in the three experimental setups, for IR-Vis SFG, PM-IRAS, and STM measurements, respectively. The Ir(111) single crystal (Mateck) was cleaned by standard cycles of $Ar^+$ sputtering and annealing in UHV, alternated with oxygen treatments. Temperature was monitored by means of a K-type thermocouple. A complete graphene single sheet was grown on Ir(111) by thermal cracking of ethylene dosed from the background in vacuum, following established procedures[26,48]. After saturation with ethylene at room temperature, the crystal was annealed to 1100 K in vacuum. At 1100 K, an ethylene background was introduced, and the temperature was further increased up to 1300 K. Temperature cycles (1300–500–1300 K) followed, always in ethylene background. In order to check the integrity of the graphene layer, CO adsorption experiments were performed. CO does not stick on a complete graphene sheet, and intercalation occurs at mbar pressure only if defects are present[26]. In the latter case intercalated CO, adsorbed at the Ir(111) surface, contributes with a very intense vibronic feature in the range 2040–2080 cm$^{-1}$ [26]. In the IR-Vis SFG spectra, only the graphene phonon could be observed: more specifically, no signal could be detected in the C–O stretching region (see Supplementary Fig. 1). Furthermore, position, phase, and intensity of the GR-related vibron did not change, thus confirming the graphene integrity. FePc molecules (TCI Europe, I0783-1G, 132-16-1, purity 98%) were evaporated from a quartz crucible, heated in UHV by thermal contact with a resistively heated tantalum filament. After an initial outgassing treatment, the molecules were evaporated from the crucible with a background pressure in the low $10^{-10}$ mbar. No SFG signal originated from the FePcs adsorbed on graphene. Indeed, their planar adsorption geometry, due to the weak coupling with the carbon sheet[15], yields no SFG active modes, i.e. vibrational modes with a dipole component parallel to the surface normal.

**Infrared-Visible Sum-frequency generation spectroscopy.** IR-Vis sum-frequency generation vibronic spectroscopy measurements were performed in a dedicated setup[32]. A UHV system with a base pressure of $5 \times 10^{-11}$ mbar hosts standard surface science preparation and characterization techniques, and it is directly connected to a high-pressure cell for in situ IR-Vis SFG spectroscopy. The reactor is equipped with a gas system to handle the reactants' pressure in the $10^{-9}$–$10^{+2}$ mbar range. CO was dosed from aluminum bottles through aluminum piping in order to avoid the formation of Ni carbonyls. The inlet and outlet of the laser beams are provided by UHV-compatible BaF$_2$ windows. The Ir(111) disc was mounted on Ta wires, used also for resistive heating. The excitation source (Ekspla) delivers a 532 nm (2.33 eV, 30 ps, 50 Hz, 0.01 mJ pulse$^{-1}$ at 1% power) visible beam and tunable IR radiation in the 1000–4500 cm$^{-1}$ range. After normalization to the IR and visible excitation intensities, the SFG spectra were analyzed by least-squares fitting to a parametric, effective expression of the nonlinear second-order susceptibility[32,49]. The expression (reported here below) well reproduces the observed lineshapes, accounting for the resonant IR-Vis vibronic transitions and for the nonresonant background, and describing all the interference terms:

$$\frac{I_{SFG}(\omega_{IR})}{I_{vis}I_{IR}(\omega_{IR})} \propto \left| A_{NRes} + \sum_k \frac{A_k e^{i\Delta\varphi_k}}{\omega_{IR} - \omega_k + i\Gamma_k} \right|^2 \qquad (1)$$

where, $A_{NRes}$ and $A_k$ account for the amplitudes of the nonresonant and $k^{th}$-resonant contributions, respectively, $\Delta\varphi_k$ is the phase difference between the $k^{th}$-resonant and nonresonant signals, $\omega_k$ is the energy position of the line, $\Gamma_k$ its Lorentzian broadening, related to the dephasing, which in turn stems from the energy lifetime and from the elastic dephasing of the excited vibronic state[50]. In order to account for inhomogeneity broadening, the lineshape was also convoluted with a Gaussian envelope. In the figures of this paper, we plot the normalized IR-Vis SFG signal intensity (gray dots) together with the best fit (blue lines). We also plot (color-filled curves) the intensity of each resonance and its interference with the nonresonant background by calculating, with the parameters obtained from the fitting procedure, the following quantity:

$$\frac{I_{SFG,k}(\omega_{IR})}{I_{vis}I_{IR}(\omega_{IR})} \propto \left| A_{NRes} + \frac{A_k e^{i\Delta\varphi_k}}{\omega_{IR} - \omega_k + i\Gamma_k} \right|^2 \qquad (2)$$

These plots directly put in evidence the amplitude and the relative phase for each of the resonances. Further details can be found in our previous work[32]. In the present study, all spectra were collected in the ppp polarization configuration (SFG-visible-infrared).

**Polarization–modulation infrared adsorption spectroscopy**. The PM-IRAS experiments were performed in a custom-built UHV experimental chamber, previously described and successfully applied in various investigations. In brief, the chamber comprises two parts, a classical UHV chamber and an UHV-compatible high-pressure cell[51,52]. The upper UHV section allows sample preparation and characterization by standard tools of surface science, such as ion bombardment and annealing, X-ray photoelectron spectroscopy (XPS), low energy electron diffraction (LEED), and temperature programmed desorption (TPD). The sample can be transferred under UHV to the lower section, a gold-coated spectroscopic high-pressure cell in which polarization–modulation IRAS (PM-IRAS) experiments can be performed in a pressure range from UHV to $10^3$ mbar. PM-IRAS (Bruker IFS 66 v/S FTIR spectrometer, liquid $N_2$-cooled HgCdTe detector, HINDS PEM-90 ZnSe photoelastic modulator) was exploited to investigate the vibrational properties of the species adsorbed on the sample. The basic principle of PM-IRAS is that the local electric field induced by s-polarized light, when impinging at near-grazing incidence on a metal surface, is vanishing[53]. Hence, no absorption of s-polarized light will occur by molecules adsorbed on the surface. On the other hand, p-polarized light will be absorbed by molecules both in the gas phase and adsorbed on the surface. Consequently, the demodulation of the signal obtained from s- and p-polarized light provides inherently surface specific information.

**Scanning tunneling microscopy**. Low Temperature Scanning Tunneling Microscopy (LT-STM) measurements were carried out at 4 K with an Omicron LT-STM system. The microscope is hosted in a UHV chamber, operating at a base pressure of $1 \times 10^{-10}$ mbar. Images were collected in constant current mode, with the bias applied to the sample and a grounded tungsten tip. STM images were analyzed by subtracting a background plane to correct the sample tilting and by applying an appropriate affine transformation to calibrate the lateral scale.

**Time-resolved two-photon photoemission**. 2PPE measurements were carried out in the SUNDYN-ANCHOR end-station of the ALOISA beamline at the Elettra synchrotron radiation facility in Trieste (Italy). Pump-and-probe pulses were the second and fourth harmonic of the output of a Yb:YAG fiber laser (Amplitude Systèmes, Tangerine HP). The sample was excited by second harmonic pulses ($h\nu$ = 2.4 eV, 515 nm) and probed by fourth harmonic pulses ($h\nu$ = 4.8 eV, 257.5 nm) of the laser. The time delay between pump-and-probe pulses was varied via a motorized delay line. The repetition rate of the laser was 577 kHz, and the pulse width was 350 fs in the fundamental harmonic. We employed pump energies of about 0.1 μJ pulse$^{-1}$. The spot diameters for the pump and the probe beams were 350 μm and 250 μm, respectively. The emitted electrons were detected at normal emission through a hemispherical electron energy analyzer (PSP Vacuum, Resolve 120) equipped with a delay-line detection system. A bias of $-1.2$ V was applied to the sample and the spectra were acquired with a pass energy of 2 eV. TR-2PPE spectra have been acquired with p-polarized pump, while s-polarized probe was used to minimize the contribution of photoelectrons exclusively emitted by the probe photons, which are not pump-dependent. The energy scale of the 2PPE spectra has been aligned to the Fermi level by measuring the onset of the secondary electron cut-off with a low laser fluence and subsequently determining the work function of the sample with a conventional He discharge lamp.

**Theoretical methods**. All calculations are performed in the framework of density functional theory (DFT)[54], as implemented in the Quantum Espresso suite[55], with a plane-wave basis set and Vanderbilt ultrasoft pseudopotentials[56]. Energy cutoffs of 35 and 280 Rydberg have been employed for wavefunctions and electron density, respectively. Exchange and correlation were described by the functional of Perdew, Burke and Ernzerhof (PBE)[57], together with a Hubbard correction[58], acting on the $3d$ states of iron. This DFT + U approach was using the formulation of Dudarev et al.[59] A Hubbard parameter $U = 3.7$ eV was chosen, in order to reproduce experimental photoemission spectra of FePc molecules by Brena et al.[60] Moreover, van der Waals interactions were described by the Grimme potential[61]. Methfessel-Paxton broadening was used for the electronic occupation[62], with a smearing of 0.02 Ry. The triplet state was simulated imposing a total magnetization of 2.0 μB. The isolated molecule was simulated in a (25 Å × 25 Å × 15 Å) cell, and integration in the first Brillouin Zone was performed by taking the Γ point only. For the array of molecules a (13.6 Å × 13.5 Å × 15 Å) cell was needed, and a (2 × 2 × 1) grid of k-points was generated by the method of Monkhorst and Pack[63]. Relaxation was performed until forces were smaller than 0.001 a.u. A more precise setup was used for the phonon calculations, including energy cutoffs of 70 and 840 Ry, and a convergence criterion for the forces of $10^{-6}$ a.u. Frozen phonon calculations were performed perturbing along the z direction Fe, C, and O only.

## Data availability
The data that support the findings of this study are available from the corresponding author upon reasonable request.

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

## Acknowledgements

A.F. would like to acknowledge Jutta Rogal for helpful discussions. E.V. acknowledges Silvio Modesti and Peter Weinberger for their precious comments. Financial support from the University of Trieste through project FRA2016 and from Beneficentia Stiftung is kindly acknowledged. M.R. and G.R. acknowledge the FWF Doctorate School DK + Solids4Fun (W1243). Computational resources were provided by ICTP. M.D.A. and R.C. acknowledge support from the SIR grant SUNDYN [Nr. RBSI14G7TL, CUP B82I15000910001] of the Italian Ministry of Education Universities and Research MIUR. M.D.A. and R.C. thank Alberto Morgante and Albano Cossaro for fruitful discussions.

## Author contributions

M.C., Z.F, M.R., C.R., G.R., G.C., M.D.A., R.C., and E.V. contributed in the experimental part of the work while A.F., M.R., G.P., and N.S. contributed in the ab initio calculations.

## Additional information

**Competing interests:** The authors declare no competing interests.

