## [Peer Review File · Nature Communications]

Reviewer #1 (Remarks to the Author):

This paper by Corva et al. describes the elaboration of an iridium/graphene/iron phthalocyanine layered systems and its reaction with carbon monoxide. The accent of this work is made on the vibronic and vibrational studies of the CO-loaded system, where the carbon dioxide is axially coordinated to complex molecules. Sum frequency generation spectroscopy revealed extra vibronic signals, that are interpreted as coming from light-induced excited states.

To my opinion, the mechanism that is proposed by authors is possible, but needs additional proofs. At least, the possibility of LIESST effect in this system should be confirmed. That is why the paper needs a major revision before it can be recommended for publication. My principal comments are summarized below.

Major points:

To prove the LIESST effect in their system, authors should do a low-temperature excitation of their CO-containing system by visible light. In such way some iron excited centers will be frozen in high-spin state and could be characterized (by PM-IRAS, LT-STM or any other available technique). Comparison of results prior and after excitation can allow to judge on the formation of excited spin states.

How does the deposition of FePc affect the IR-Vis SFG signal? What is the effect of CO uptake? First, the deposition may influence on the GP layer properties. On the other hand, the LIESST-related deformations would also influence the GP signal.

Can authors detect IR signal from FePc molecules before and after CO uptake (e.g., some aromatic stretching modes)? Bands from organic ligands must also be sensitive to spin transitions in the iron centre.

From the manuscript, one can hardly understand on the role of localized spin excitons in all described processes. Spin excitons are postulated in the article name and explicitly discussed in the introduction part, and even appear in the conclusions. But no information can be found in result and discussion parts. When you tell long-lived excitons, what life time do you mean? Can it be calculated?

Minor points:

What is already known about the reaction between iron(II) phthalocyanine and carbon monoxide? There is no much discussion on this topic. Can one isolate the product of monocarbonylation? What spin state does it have? Are there experimental evidences for S=0? Is double carbonylation practically possible? All these points should be discussed in the manuscript.

Authors should show spectra from fig. S1 in the larger region. CO molecules that do not participate in coordination bonding can have stretching above 2100 cm⁻¹. However, only the region where metal-coordinated CO molecules can be detected is shown. If this spectral information is not available, one should comment why intercalation is not possible without coordination to the metal.

Except experimental data point and ab initio simulation, the fitting curve should be provided in fig. 2.

Reviewer #2 (Remarks to the Author):

The manuscript by Corva et al presents a spectroscopic Vibrational-SFG / IR study of 2D metalorganic crystals. Authors see the difference in the vibrational responses detected by both techniques and interpret the difference as an effect of excited states created (as a side product) in V-SFG experiments. In general, I like the idea of V-SFG detected excited states and enjoyed reading the paper.

However, I have serious concerns about the general interest of this work and about the data interpretation.

I believe the paper might be suitable for publication in a more specialised journal after doing control experiments and a major revision.

My main concerns are summarised below:

1) SFG is sensitive to both IR and Raman active modes. Therefore a simple comparison between SFG and IR spectra does not provide an evidence for new vibrational modes present in the system. Also SFG has much more complex dependence of signal on beam polarisations and molecular orientation. To convincingly show SFG addresses new vibrations (arguably belonging to the excited states) Authors would need to build a complete experimental-data-based model of IR and Raman active vibrations in the studied region and evaluate how the SFG/IR responses of these vibrations might depend on the molecule orientation and probe beams' polarisations.

2) Authors do not perform an essential control experiment, where SFG is measured without resonant electronic excitation. One obvious option might be doing measurement with 800nm VIS beam which (I might be wrong) is not resonant with the electronic transition of the molecule. I note that even this experiment should be done and analysed very carefully as even lower-frequency VIS light will be resonant with graphene or might induce multiphoton excitation.

3) Taking 1 and 2 are carefully addressed, the paper might introduce a new technique suitable for local probing of 2D excitons. That would be a very substantial achievement suitable for publication in a good specialised journal. However, without additional work it is not clear what kind of fundamental insight about material properties can be revealed by V-SFG exciton probes. I think this diminishes the paper's attractiveness for the broad readership of Nature Commun.

Answers to the Reviewers and Changes introduced into the Manuscript

Reviewer 1

1. *This paper by Corva et al. describes the elaboration of an iridium/graphene/iron phthalocyanine layered systems and its reaction with carbon monoxide. The accent of this work is made on the vibronic and vibrational studies of the CO-loaded system, where the carbon dioxide is axially coordinated to complex molecules. Sum frequency generation spectroscopy revealed extra vibronic signals, that are interpreted as coming from light-induced excited states. To my opinion, the mechanism that is proposed by authors is possible, but needs additional proofs. At least, the possibility of LIESST effect in this system should be confirmed. That is why the paper needs a major revision before it can be recommended for publication. My principal comments are summarized below. Major points: To prove the LIESST effect in their system, authors should do a low-temperature excitation of their CO-containing system by visible light. In such way some iron excited centers will be frozen in high-spin state and could be characterized (by PM-IRAS, LT-STM or any other available technique). Comparison of results prior and after excitation can allow to judge on the formation of excited spin states.*

We thank the referee for raising this point and for suggesting that a more direct experimental evidence was necessary to support our interpretation of the results. Therefore, instead of performing low-temperature experiments in order to “freeze” the excited states, as suggested, or to scavenge through the origin of the SFG signal, we have pursued an independent experimental proof. This required a considerable effort, but yielded deeper insight into the electronic mechanisms involved in the process. We exploited a state-of-the-art two-photon pump-probe, laser-based approach (2-photon photoemission spectroscopy – 2PPE). Therefore, we investigated the evolution of the excited states with a time resolution of the order of 100 fs and at room temperature, thus without slowing down the dynamics. In this way, we were able to follow the evolution of the electronic excited states. We are glad to report that we obtained the experimental, direct evidence that the mechanism we originally proposed is the right one, thus involving a short-lived state ($\tau < 600$ fs) that decays in a long-lived state ($\tau = 28 \pm 8$ ps). Moreover, the associated energies obtained from these new data are 1.3 and < 0.6 eV, compatibly with a singlet fission process, since the energy of the second state is less than half the energy of the first state. We now report these important additional results in the manuscript. We also added a dedicated section and a new Figure in the Supporting Information section.

2. *How does the deposition of FePc affect the IR-Vis SFG signal? What is the effect of CO uptake? First, the deposition may influence on the GP layer properties. On the other hand, the LIESST-related deformations would also influence the GP signal. Can authors detect IR signal from FePc molecules before and after CO uptake (e.g., some aromatic stretching modes)? Bands from organic ligands must also be sensitive to spin transitions in the iron centre.*

We thank the reviewer for this comment. Unfortunately, we were unable to detect signals from the FePc intramolecular vibrational modes. As reported in the literature [*J. Chem. Phys.* 141, 184308 (2014)], this is explained by the weak coupling of the Pc molecules with the underlying graphene, yielding an almost planar geometry of the molecule. Thus, no dipolar modes with a significant component parallel to the surface normal are present, yielding no SFG, and neither IR active modes, at variance with the case of a direct FePc-to-metal bond [*J. Phys. Chem. C* 120, 22298 (2016)]. As a consequence, FePc molecules

on graphene/Ir(111) are “invisible” to IR-based probes. We now write this in the Supporting Information section on page 2.

3. *From the manuscript, one can hardly understand on the role of localized spin excitons in all described processes. Spin excitons are postulated in the article name and explicitly discussed in the introduction part, and even appear in the conclusions. But no information can be found in result and discussion parts.*

We thank the referee for this observation. Indeed, the definition “spin excitons” that we adopted in the original submission was misleading. While we aimed at addressing the electronic excitation, combined with a change in the single-molecule spin in the singlet fission mechanism, we recognized that the definition “spin-exciton” typically refers to collective spin excitations in magnetic materials [see for example J. Phys. Conf. Series 391 (2012) 012043]. Instead, the use we made of the term originates from Nat. Chem. 8 (2016) 16, reporting molecular excitons. Nevertheless, in order to avoid possible confusion, we now refer only to excitons and describe the spin configurations separately. The title and the text have been modified accordingly.

4. *When you tell long-lived excitons, what life time do you mean? Can it be calculated?*

Again, we really thank the reviewer for this comment. We have now measured the lifetimes (28 ± 8 ps at room temperature). See answer to Question 1 for further details. The text has been modified accordingly and new data have been added in the Supporting Information section.

5. *Minor Points: What is already known about the reaction between iron(II) phthalocyanine and carbon monoxide? There is no much discussion on this topic. Can one isolate the product of monocarbonylation? What spin state does it have? Are there experimental evidences for $S=0$? Is double carbonylation practically possible? All these points should be discussed in the manuscript.*

We thank the referee for these comments. As reported in the text, the FePc displays a $s = 1$ configuration that is quenched to $s = 0$ upon carbonylation. Double carbonylation is found to be energetically not possible, at least in a cys configuration as upon adsorption on a supporting surface. The spin state has already been measured, for a review see J. Phys. Condens. Matter. 29, 23001 (2017) and Surf. Sci. Rep. 70, 259 (2015), which are cited in the text. A specific case of spin quenching of the FePc induced by CO adsorption from $s = 1$ to $s = 0$ is reported in the literature in J. Phys. Condens. Matter. 22, 472002 (2010), which we now cite in our manuscript in response to the reviewer comment.

6. *Authors should show spectra from fig. S1 in the larger region. CO molecules that do not participate in coordination bonding can have stretching above 2100 cm^{-1} . However, only the region where metalcoordinated CO molecules can be detected is shown. If this spectral information is not available, one should comment why intercalation is not possible without coordination to the metal.*

Thanking the reviewer for this point, we remark that CO intercalation under the GR sheet is not allowed in the near-ambient pressure range, as reported in the literature [ACS Nano 11, 1041 (2017)]. The CO/Ir(111) stretching mode is expected in the 2040-2080 cm^{-1} range and is therefore excluded. Due to the nature of the CO-Ir bond (strongly exothermic and non-activated), molecules intercalating under the GR sheet would promptly bind and stabilize at the underlying metal surface at room temperature. Additional proof is provided by the GR vibron in Fig. S1, which does not shift in energy, nor changes its phase and intensity upon exposure of GR to CO. The latter quantities would be indeed strongly

affected by CO intercalation. This is now explained in the Supporting Information section on page 2.

7. *Except experimental data point and ab initio simulation, the fitting curve should be provided in fig. 2.*

We thank the reviewer, we now plot in Fig. 2c also the Langmuir model fitting curve, yielding a CO-Fe binding energy of 0.3 eV.

Reviewer 2

1. *The manuscript by Corva et al presents a spectroscopic Vibrational-SFG / IR study of 2D metalorganic crystals. Authors see the difference in the vibrational responses detected by both techniques and interpret the difference as an effect of excited states created (as a side product) in V-SFG experiments. In general, I like the idea of V-SFG detected excited states and enjoyed reading the paper. However, I have serious concerns about the general interest of this work and about the data interpretation. I believe the paper might be suitable for publication in a more specialised journal after doing control experiments and a major revision. My main concerns are summarised below: SFG is sensitive to both IR and Raman active modes. Therefore a simple comparison between SFG and IR spectra does not provide an evidence for new vibrational modes present in the system. Also SFG has much more complex dependence of signal on beam polarisations and molecular orientation. To convincingly show SFG addresses new vibrations (arguably belonging to the excited states) Authors would need to build a complete experimental-data-based model of IR and Raman active vibrations in the studied region and evaluate how the SFG/IR responses of these vibrations might depend on the molecule orientation and probe beams' polarisations.*

We thank the reviewer for raising this point. Since the selection rules for the SFG process include both IR (vibrational) and Raman (electronic) matrix elements, a transition generating a peak in IR spectroscopy may be forbidden in SFG, but not vice versa. This is, indeed, why we consider the comparison between IR and IR-Vis SFG data sets as a proof of the electronic origin of the multiplet. In addition, by changing the polarization of the IR, Vis, and signal beams in the SFG experiments we could not detect any signal. This is consistent with the subset of elements of the susceptibility tensor that are sampled in the case of molecules adsorbed at a metal surface and with a C-O bond parallel or almost parallel to the surface normal.

2. *Authors do not perform an essential control experiment, where SFG is measured without resonant electronic excitation. One obvious option might be doing measurement with 800nm VIS beam which (i might be wrong) is not resonant with the electronic transition of the molecule. I note that even this experiment should be done and analysed very carefully as even lower-frequency VIS light will be resonant with graphene or might induce multiphoton excitation.*

The referee is right, and she/he is also right when stating that even a lower-frequency light would be resonant with the system. For this reason, recognizing that this issue was relevant, we decided to pursue an independent experimental proof supporting our interpretation of the electronic transitions involved in the process. Thus, we focused on the VIS light absorption. This required a considerable effort, but yielded deeper insight into the electronic mechanisms involved in the process. We exploited a state-of-the-art two-photon pump-probe, laser-based approach (2-photon photoemission spectroscopy – 2PPE). So, we investigated the excited state properties with a time resolution of the order of 100 fs and at

room temperature. In this way, we were able to follow the time evolution of the electronic excited states. We are glad to report that we obtained the experimental, direct evidence that our proposed mechanism was the right one, thus involving a short-lived state ($\tau < 600$ fs) that decays in a long-lived state ($\tau = 28 \pm 8$ ps). Moreover, the associated energies obtained from these new data are 1.3 and < 0.6 eV, compatibly with a singlet fission process, since the energy of the second state is less than half the energy of the first state. We now report these additional results in the manuscript. We also added a dedicated section and a new Figure in the Supporting Information section with details about the experimental setup.

3. *Taking 1 and 2 are carefully addressed, the paper might introduce a new technique suitable for local probing of 2D excitons. That would be a very substantial achievement suitable for publication in a good specialised journal. However, without additional work it is not clear what kind of fundamental insight about material properties can be revealed by V-SFG exciton probes. I think this diminishes the paper's attractiveness for the broad readership of Nature Commun.*

We fully agreed with the referee and therefore performed additional measurements focusing on the electronic transitions involved in the process. See Point 2 for further details.

Reviewer #1 (Remarks to the Author):

Authors considered all remarks made in my previous review. The manuscript was considerably improved after adding new TR-2PPE results. I can now recommend the publication of this paper.

Reviewer #2 (Remarks to the Author):

The revised version of the manuscript does not address my criticism. I was questioning the validity of interpretation and requested/proposed the following two additions to the study:

- the IR/Raman mode analysis of contributions to the observed SFG modes and spectroscopic model of system responses (NOT PERFORMED)
- control experiments with non-resonant excitation to verify the nature of SFG signals (NOT PERFORMED)

This leave conclusions very questionable and I dont think allow manuscript's publication.

After reading paper ones again, I would like to add two new points to my previous review:

- The new photo-emission study is interesting but to my mind the very short excitonic lifetimes observed are in the conflict to the idea of long-lived and well controlled excitons promised in introduction. The system authors study just relaxes too fast to be interesting for applications.
- The paper is in general poorly written (I was probably too fast reading it first time). Particularly introduction is messy (poor language and structure, mixing up coherence and lifetime of excitons, attempting to give a length discussion of singlet fission that is irrelevant to the subject). I got a feeling that authors just trying to mention as many hot topics and fashionable terms as possible, but those are not really relevant to the studied systems.

In summary, potentially an interesting 'technique demonstration' work but with clear methodological faults. Taking limited impact and poor presentation the paper is not suitable for Nature Comm.

Reply to the comments of Reviewer 2.

- 1) *“The revised version of the manuscript does not address my criticism. I was questioning the validity of interpretation and requested/proposed the following two additions to the study: the IR/Raman mode analysis of contributions to the observed SFG modes and spectroscopic model of system responses (NOT PERFORMED)”*: we did not perform this estimate specifically because we made a more thorough computational, *ab initio* modeling of the whole system, calculating phonons and comparing the results with our experiments, thus going much deeper in the mechanisms than what would have been achieved by only modeling the IR/Raman modes, as suggested by the referee; furthermore, the subject of interest is only the internal C-O stretching mode, whose energy range is well-known in the literature. In the present case the fine structure of the mode has been calculated by means of state-of-the-art approaches since electronic excited states had to be accounted for in the computation, a non-trivial task for ground state density functional theory.
- 2) *“Control experiments with non-resonant excitation to verify the nature of SFG signals (NOT PERFORMED)”*: we agree with the referee that a doubly-resonant SFG experiment would have been interesting. This, though, would have required accessing a different expensive setup with a different excitation wavelength, which was not available to us; in any case, we did much more than that; indeed, we measured the system with no excitation at all (PM-IRAS measurements, fig. 4 of the manuscript), thus demonstrating beyond any doubt what the referee was asking for.
- 3) *“After reading paper ones again, I would like to add two new points to my previous review: The new photo-emission study is interesting but to my mind the very short excitonic lifetimes observed are in the conflict to the idea of long-lived and well controlled excitons promised in introduction. The system authors study just relaxes too fast to be interesting for applications”*: we have provided a measurement of the lifetime (28 ± 8 ps), which is in line with other findings on model systems considered relevant, in perspective, for applicative organic devices, where tens or hundreds of picoseconds account for “long” lifetimes, at variance with “short” ones corresponding to tens or hundreds of femtoseconds. See for example: Nature Communications 4 (2013) 2679 (45 ps), Nature Chemistry 4 (2012) 840 (70 ps), Nature Chemistry 9 (2017) 341 (30 ps), Nature Materials 14 (2015) 426 (from 13 to 80 ps), and the already cited Science 334 (2011) 1541 (5 ps and 250-350 ps). We now state this with clarity by citing all these reference papers in the manuscript [former ref 42, and refs 46-49].
- 4) *“The paper is in general poorly written (I was probably too fast reading it first time). Particularly introduction is messy (poor language and structure, mixing up coherence and lifetime of excitons, attempting to give a length discussion of singlet fission that is irrelevant to the subject). I got a feeling that authors just trying to mention as many hot topics and fashionable terms as possible, but those are not really relevant to the studied systems.”*: We were surprised by this comment on the quality of the manuscript since this referee formerly explicitly appreciated the paper style in his/her first review: *“I enjoyed reading the paper”*. As to the relevance of the singlet fission to the manuscript topic we disagree with the referee’s opinion. To give just a first straightforward example, *“Harvesting singlet fission for solar energy conversion via triplet energy transfer”* is the title of a Nature Communication paper (4, 2013, 2679) regarding CuPc molecules. Singlet fission is indeed the core mechanism explaining what we observed and it is widely

accepted as the very basis of modern applicative ideas of two-for-one light harvesting devices. This is the reason why it is at present a very hot research topic, attracting the attention of top journals, as shown above.

Reviewer #3 (Remarks to the Author):

I am convinced that the authors can see excited states on the chromophore via monitoring the vibrational modes of CO. I think this is sufficient for publication in Nat. Comm.. I am not convinced that set up is suitable for quantum computation but in my view this is not essential at this stage.

The introduction is a bit unfocused. I can understand it sentence by sentence but the direction is not clear. I would suggest shortening and increase the focus on what is relevant here.

The system made by the authors is beautiful and an excellent playground for fundamental physics studies that may suggest new ideas to others.

There is some mismatch between the complex interpretation and limited data available with ab initio calculations used to fill the gap. This is the main reason why I am not sure more complex control of the excited state dynamics will be possible.

The sentence "2PPE experiments associate the two excited states with energies of 1.3 and <0.6 eV, respectively." Is missing a reference which seems quite critical in this case.

Answers to the Reviewer and Changes introduced into the Manuscript

Reviewer 3

- 1. I am convinced that the authors can see excited states on the chromophore via monitoring the vibrational modes of CO. I think this is sufficient for publication in Nat. Comm.. I am not convinced that set up is suitable for quantum computation but in my view this is not essential at this stage. The introduction is a bit unfocused. I can understand it sentence by sentence but the direction is not clear. I would suggest shortening and increase the focus on what is relevant here. We agree with the reviewer that the introduction was a bit long, making reference to a huge number of examples and to many mechanisms related with exciton creation and time evolution. In particular, the discussion about the hemozoin polymer and the Fe(III) complex was too uncorrelated with the specific focus of the paper, thus defocusing the introduction. Thus, we cut that part (together with former references 9 and 10), and directly focused on the exciton dynamics that is the main subject of our work.*
- 2. The system made by the authors is beautiful and an excellent playground for fundamental physics studies that may suggest new ideas to others. There is some mismatch between the complex interpretation and limited data available with ab initio calculations used to fill the gap. This is the main reason why I am not sure more complex control of the excited state dynamics will be possible. The sentence "2PPE experiments associate the two excited states with energies of 1.3 and <0.6 eV, respectively." Is missing a reference which seems quite critical in this case. We thank the referee for raising this issue. Indeed, the sentence*

was not clear, since the values are our results and not data from the literature. The measurement details are reported in the supplementary information. We now changed the sentence to: “Moreover, we find energies of 1.3 and < 0.6 eV for the two excited states, respectively” and added explicit reference to the supplementary information section.